# LILE: Look In-Depth before Looking Elsewhere – A Dual Attention Network using Transformers for Cross-Modal Information Retrieval in Histopathology Archives

**Danial Maleki**[1]                                     DMALEKI@UWATERLOO.CA

**H.R Tizhoosh**[1,2,†]                             TIZHOOSH.HAMID@MAYO.EDU

[1] *Kimia Lab, University of Waterloo, Waterloo, ON, Canada*

[2] *Artificial Intelligence and Informatics, Mayo Clinic, Rochester, MN, USA*

† *Corresponding author*

## Abstract

The volume of available data has grown dramatically in recent years in many applications. Furthermore, the age of networks that used multiple modalities separately has practically ended. Therefore, enabling bidirectional cross-modality data retrieval capable of processing has become a requirement for many domains and disciplines of research. This is especially true in the medical field, as data comes in a multitude of types, including various types of images and reports as well as molecular data. Most contemporary works apply cross attention to highlight the essential elements of an image or text in relation to the other modalities and try to match them together. However, regardless of their importance in their own modality, these approaches usually consider features of each modality equally. In this study, self-attention as an additional loss term will be proposed to enrich the internal representation provided into the cross attention module. This work suggests a novel architecture with a new loss term to help represent images and texts in the joint latent space. Experiment results on two benchmark datasets, i.e. MS-COCO and ARCH, show the effectiveness of the proposed method.

**Keywords:** Cross-Modal Retrieval, Histopathology, Attention

## 1. Introduction

A plethora of real-world applications is emerging due to the rapid expansion in the volume and variety of data on the one hand and the recent success of deep learning on the other hand. However, the majority of existing models can only work with a single modality of data, e.g., only analyzing images or text. This restriction on the data source prevents the models from having a more generalizable and robust problem representation for downstream tasks. The absence of a model that can be used across several modalities and that can use information from multiple sources of data is in great demand and has become increasingly common in both academia and industry. One task that has gained popularity over the years is retrieving the relation between one of the data modalities; a task called "cross-modality retrieval". This task seeks to bring a pair of data from two distinct sources together by recognizing their intrinsic relationship. Because such cross-modal topologies need to learn the inter-modal correspondence and modality representation separately, their design and training can be rather challenging. Image and text are the most extensively utilized and commonly observed modalities in real-world data. However, The diversity in information sources can be observed more commonly in the medical field. Images, reports and molecular

data in patients' medical records are different and indispensable sources of information. A model able to retrieve a source of data when a different one is available can help provide complete information about the patient.

The common solution to approach to explore the relationship between image and text is to map the visual semantic embeddings (Karpathy and Fei-Fei, 2015; Frome et al.) of an image and the corresponding words/phrases/sentences into a common latent embedding space (Barnard et al., 2003; Berg et al., 2004; Socher and Fei-Fei, 2010; Chong et al., 2009). In these methods, the goal is generally to find a common space where the corresponding representations of images and text are as close as possible, hence making recognizing their relationship easier. Recent studies investigate the use of the attention mechanism to understand the similarity between the two modalities. The majority of studies in this category have employed cross-attention mechanisms, which allow the model to selectively attend to the parts of an instance that are relevant to the context from the other modal (Lee et al., 2018; Huang et al., 2017). Other methods attempt to refine their representation regarding the information of the other modal (Chen et al., 2020). Nonetheless, due to the gap between the representation of images and texts, these methods may not find the optimal representation. Moreover, ambiguity in text documents is a common challenge posing a learning obstacle to the model if it uses an injective embedding. Aside from that, humans use a hierarchical structure to organize and store diverse semantic concepts. However, the majority of the currently available approaches group semantics together in a consistent manner (Lee et al., 2018; Chen et al., 2020; Song and Soleymani, 2019).

In this study, the mentioned issues are addressed by providing an iterative regime to capture related information in the other modality and extract the most significant features by looking into its context. It can help the model capture the information related to the other modality and itself simultaneously. Accordingly, such an approach can help extract richer latent embeddings for each instance. Moreover, training the model in an iterative regime can help the model to capture higher-level features gradually based on the features acquired in the previous steps. Furthermore, refining the extracted features based on the intersection between modalities can help the model to enrich its representations.

## 2. Literature Review

Many works on cross-modality retrieval have been looking for a similar pattern in the feature representation of both modalities (Karpathy and Fei-Fei, 2015; Lee et al., 2018; Huang et al., 2018; Lu et al., 2020; Radford et al., 2021). These studies can generally be categorized into global and local feature matching approaches.

**Global feature matching methods** were among first attempts to solve the image-text retrieval task. Kiros *et al.* (Kiros et al., 2014) proposed to use a CNN and a RNN to encode given images and text data. For the loss function, they proposed an idea to implement a pairwise ranking loss objective to rank images and correspondence descriptions. Faghri *et al.* (Faghri et al., 2017) modified the loss function with hard negatives in the triplet loss function and applied on a similar architecture to (Kiros et al., 2014). Furthermore, by improving the quality of generative models (Zhu et al., 2017; Karras et al., 2019), some studies have attempted to exploit the capability of generative models to enhance the performance of proposed methods (Patrick et al., 2020; Peng and Qi, 2019; Gu et al., 2018).

**Local feature matching methods** consider the correspondence between image and text feature representations at the level of image regions and words. Karpathy *et al.* (Karpathy and Fei-Fei, 2015) proposed a method to extract features for image regions at object level using Faster R-CNN (Ren et al., 2015) and text words, and then align them into a common space. However, as the application of the attention mechanism in a variety of tasks has become more common, several approaches in image-text retrieval seek to leverage the attention mechanism (Chen et al., 2020; Lee et al., 2018; Song and Soleymani, 2019; Thomas and Kovashka, 2020; Lee et al., 2018). Yale *et al.* (Song and Soleymani, 2019) proposed a method to combine global features with locally-guided features via a multi-head self-attention module. Christopher *et al.* (Thomas and Kovashka, 2020) in a similar network proposed a novel within-modality loss function that drives feature representation toward more coherence. The majority of these methods rely on the premise that vision and text are mutually exclusive and equally important. However, grounding (or base) representation of each modality derived from another modality to finer details is critical for bridging the gap between vision and text modality. Kuang-Huei (Lee et al., 2018) was one of the first studies that investigated the use of cross-attention to explore full latent matching using image regions and words in the text as context.

Furthermore, Hui *et al.* in IMRAM paper (Chen et al., 2020) presented an iterative alignment method that captures the fine-grained correspondence between image and text progressively using a cross-attention module. By the development of Transformer models, serious approaches were proposed for cross-modality retrieval task using Transformers (Lu et al., 2019; Tan and Bansal, 2019). Authors in (Lu et al., 2019) proposed a network architecture that consists of two single-modal Transformer networks applied on input images and texts data respectively.

Recent studies have shown that training on large-scale datasets could help models to achieve significant improvement in image-text retrieval task (Li et al., 2020; Radford et al., 2021; Jia et al., 2021). Jia *et al.* in ALIGN paper (Jia et al., 2021) leveraged a dataset of over one billion paired image and alt-text data and claimed that a simple dual-encoder network could learn to align image and text representation using a contrastive loss. Similar to ALIGN paper, Xiujun *et al.* in Oscar paper (Li et al., 2020) collected a dataset over 6.5 million paired of image and text to train their network. Another example for boosting the performance using a large amount of data can be CLIP (Radford et al., 2021) paper. Authors in CLIP paper also demonstrated that the task of image-text retrieval could be outperformed using a massive dataset, 400 million in this case. However, a bottleneck of these large-dataset-dependent approaches emerges in those domains which are unlikely to be included in their collected dataset, such as histopathology domain. Moreover, training on that amount of data requires huge computational resources.

Cross-Modality Retrieval is new to the field of histopathology. There are not many studies that explore retrieving a description correspondence to an image or vice versa. As a result, a few works were trying to retrieve or utilize image-text modalities in their suggested methods. Zhang *et al.* (Zhang et al., 2017) was among the first authors who proposed a model and dataset for image-text in pathology. Authors proposed a new framework, namely MDNet, to establish a direct multi-modal mapping between images and diagnostic reports. Moreover, they developed a private patch-based dataset based on pathology bladder cancer images. For descriptions, they asked pathologists to write 5 short sentences for each patch.

The proposed method is capable of generating short diagnostic reports and retrieving images based on symptom description. Yet the developed dataset was too small, private and limited to only H&E images. Recently, Gamper *et al.* (Gamper and Rajpoot, 2021) developed a new publicly available dataset, ARCH, that contains more than 7,000 patch-based images with the corresponded captions. The authors of the paper used PubMed medical articles (Pub) and academic pathology textbooks to collect the ARCH dataset.

## 3. Methodology

The proposed model, as demonstrated in Figure 1 includes different components. LILE, a dual attention network using Transformers, takes images and texts as inputs and extracts feature representations for each of those using Transformers. Then, a self-attention module (Vaswani et al., 2017), which is applied for extracting the most significant parts of each modality with respect to itself, is applied. For the next step, a cross-attention module and gated memory block are applied to help the model refine the representation of each instance with respect to the outputs of attention modules. Additionally, an iterative matching scheme using a gated memory block is applied to refine the extracted features for each modality.

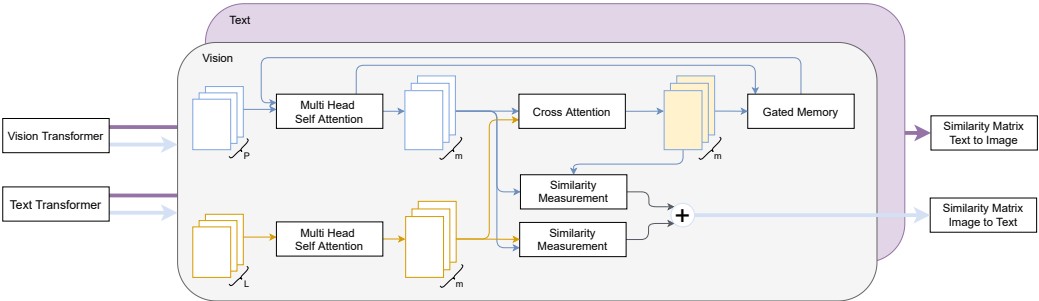

Figure 1: The architecture of LILE for cross-modality retrieval

### 3.1. Feature Representation

Data representation, or representation learning (Bengio et al., 2013), is a collection of approaches in ML that allows a model to automatically discover the representations required from the given data. Recently, Transforms become more prevalent for representation learning. the Transformers' modular architecture enables the processing of different modalities (e.g., images, videos, text, and voice) leveraging similar processing blocks. It scales efficiently to huge capacity networks for complex tasks and performs well with massive datasets. Transformer architecture is chosen for text and image feature extractors in this study due to these advantages.

**Image Representation:** In this study, a pre-trained ViT (Dosovitskiy et al., 2020) is implemented to encode input images . Each image is split into a sequence of fixed-size, non-overlapping image patches before being fed to the ViT. Utilizing object detection models is another method for extracting visual information from an image. These models are trained on the input image in order to extract the objects. As a result, the input image can be represented as a set of extracted feature maps for all of the objects in the image (Lee et al.,

2018; Chen et al., 2020). Depending on the dataset and the availability of annotations for the object detection task, one of these approaches is used in this study.

**Text Representation:** Text representation can be defined as a method for encoding sentences into vectors. Transformers have recently emerged as the top-performing solution for the majority of NLP tasks and outperform many other approaches (Liu et al., 2019; Radford et al., 2019; Usi). In this study, a pre-trained "Roberta" architecture (Liu et al., 2019) which is a Transform-based model is deployed to encode the input text. It receives a text description as an input and returns a feature map that represents the given data.

### 3.2. Attention and Gated Memory Blocks

After the representation for each modality instance has been extracted, a multi-head self-attention module is applied to obtain $m$ enhanced feature maps for extracted features from the previous stage similar to the (Wei et al., 2020). These $m$ representations highlight the most significant features of each modality in relation to itself.

A cross-attention mechanism is utilized to attend to distinct parts of one modality given the context of another modality as proposed by authors in (Chen et al., 2020; Wei et al., 2020). The application of cross-attention in the proposed approach is to attend to image regions regarding the text input tokens and vice versa. The gated memory block seeks to refine the extracted features from each modality considering the cross-attention feature maps. The input modalities are denoted as $X = \{x_i | i \in [1, m], x_i \in \mathbb{R}^d\}$ and $Y = \{y_i | i \in [1, m], y_i \in \mathbb{R}^d\}$ which $X$ and $Y$ can be either image or text features, where $m$ is the number of attention heads in the multi-head self-attention module in the previous step. The cross-attention module helps to have the highlighted information for modality $X$ related to modality $Y$. To achieve this goal, a suitable measurement is needed to quantify the similarity between each feature map and feature maps in another modality.

In the suggested solution, the cosine similarity is applied (Ji et al., 2019; Jia et al., 2021; Lee et al., 2018). The similarity between each instance in modality $X$ and $Y$ is determined as Eq.1, where $s_{ij}$ denotes the similarity between the $i$-th feature map for modality X and the $j$-th from modality Y. Furthermore, it is beneficial to threshold the similarity at 0 and normalize it (Karpathy et al., 2014; Lee et al., 2018).

$$s_{ij} = \frac{x_i^T y_j}{||x_i|| \times ||y_j||}, \forall i \in [1, m], \forall j \in [1, m] \quad \bar{s_{ij}} = \frac{\max(0, s_{ij})}{\sqrt{\sum_{i=1}^m \max(s_{ij})^2}}. \tag{1}$$

To attend on set $Y$ with respect to a given feature $x_i$ in $X$, a weighted combination of $y_j$ is defined. The definition of attention function in Eq.1 is a variant of the "dot product attention" that is commonly used in other studies (Luong et al., 2015):

$$a_i^x = \sum_{j=1}^m \alpha_{ij} y_j, \quad \alpha_{ij} = \frac{\exp(\lambda \bar{s_{ij}})}{\sum_{j=1}^m \exp(\lambda \bar{s_{ij}})}. \tag{2}$$

In Eq.2, the parameter $\lambda$ is the inverse temperature of the softmax function (Chorowski et al., 2015). A more smooth attention function can be achieved by adjusting $\lambda$. $A^x$ is defined as $\{a_i^x \in [1, m], a_i^x \in \mathbb{R}^d\}$ where each element of $A^x$ captures significant parts of each $x_i$ given the whole $Y$ set as context. To refine the extracted features of $X$ regarding the important parts of each $x_i$ given $Y$, a memory unit has been used. It would dynamically

update and refine the feature maps of $X$ by looking to both $A^x$ and $X$ as Eq.3, where $f(\cdot)$ can be defined differently (Kalra et al., 2020; Weston et al., 2014; Burtsev et al., 2020). In this study, a gated mechanism has been adopted for $f(\cdot)$ as follows:

$$x^* = f(x_i, a_i^x). \tag{3}$$

### 3.3. Iterative Matching

As stated in section 3.2, gated memory can assist the model in refining the feature representation regarding the shared information between two modalities as proposed by authors in (Chen et al., 2020). Having the gated memory in an iterative scheme can be beneficial as each iteration step, with the help of $A^x$ feature representation of $X$ can be re-calibrated. The iterative scheme can be summarized as Eq.4, where $k$ is the iteration step that will be performed to refine the alignment for the next iteration:

$$X_k^* = \mathbf{Memory}(X_{k-1}, A^x). \tag{4}$$

At iteration step $k$, the similarity score between image I and text T is calculated as follows:

$$S_k(I,T) = \alpha \left( \frac{1}{m} \sum_{i=1}^{m} S_k^{(v,v\rightarrow T)}(v_i, T) + \frac{1}{m} \sum_{i=1}^{m} S_k^{(w,w\rightarrow I)}(I, w_i) \right) + \\ (1-\alpha) \left( \frac{1}{m} \sum_{i=1}^{m} S_k^{(v,T)}(v_i, T) + \frac{1}{m} \sum_{i=1}^{m} S_k^{(w,I)}(I, w_i) \right) \tag{5}$$

where $\alpha$ is a learnable scalar weight parameter that balances the influence of the similarity score terms. $S^{(v,v\rightarrow T)}(v_i, T)$ and $S^{(w,w\rightarrow I)}(I, w_i)$ are defined as similarity score between image regions and text $T$ and text tokens and image $I$ respectively proposed by authors in (Chen et al., 2020) . These similarity scores are derived as

$$S_k^{(v,v\rightarrow T)}(v_i, T) = \mathbf{sim}(v_i, A_k^v), \quad S_k^{(w,w\rightarrow I)}(I, w_i) = \mathbf{sim}(A_k^t, w_i) \tag{6}$$

The similarity score can be boosted by including directly the similarity between image and text as $S^{(v,T)}(v_i, T) = \mathbf{sim}(v_i, T)$ and $S^{(w,I)}(I, w_i) = \mathbf{sim}(I, w_i)$.

This can assist the model in preserving the semantic meaning of each instance while it attempts to bring paired instances closer together. To put all $k$ steps together, the similarity score between image I and text T will be derived as Eq.7, where $k$ is the number of matching steps that will be set as a hyper-parameter. See appendix D.

$$S(I,T) = \sum_{k=1}^{K} S_k(I,T). \tag{7}$$

### 3.4. Loss Function

In this study, N-pairs bi-directional triplet-loss is implemented.

$$\mathcal{L} = [\Delta - S(I_i, T_i) + S(I_i, T_j)]_+ + [\Delta - S(I_i, T_i) + S(I_j, T_i)]_+. \tag{8}$$

In Eq.8, $\Delta$ is a margin value and $[x]_+ = max(0, x)$. The term $S(I, T)$ is defined in Eq.7 and measures the similarity between image $I$ and text $T$. This similarity score forms a similarity score matrix $S$ of size $n \times n$, where $S$ is symmetric, See appendix A. In the training phase, $n$ is the size of mini-batch. Images and text with a same subscript are paired instances which means the diagonal of the matrix should have the largest value compared to the other indices. This regime will be trained in an end-to-end manner.

## 4. Experiments

We used two benchmark datasets to evaluate the effectiveness of the proposed method:

- **MS-COCO** (Lin et al., 2014), a large-scale object detection, segmentation, key-point detection, and captioning dataset, and

- **ARCH** (Gamper and Rajpoot, 2021), a computational pathology multiple-instance captioning dataset, see appendix B.

To compare the proposed method with other approaches, Recall at K (R@K) is used, which measures the fraction of queries retrieved correctly among top $K$ search results(Chen et al., 2020; Lee et al., 2018). To show the efficiency of the proposed method, the reported results also includes "R@sum" which is the summation of evaluation at different $K$, i.e., R@1+R@5+R@10 as in (Huang et al., 2017).

Table 1: Comparison with state-of-the-art methods on MS-COCO

| Method | Text Retrieval | | | Image Retrieval | | | R@sum |
|---|---|---|---|---|---|---|---|
| | R@1 | R@5 | R@10 | R@1 | R@5 | R@10 | |
| 1K | | | | | | | |
| SCO(Huang et al., 2018) | 69.9 | 92.9 | 97.5 | 56.7 | 87.5 | 94.8 | 499.3 |
| SCAN(Lee et al., 2018) | 72.7 | 94.8 | 98.4 | 58.8 | 88.4 | 94.8 | 507.9 |
| PVSE(Song and Soleymani, 2019) | 69.2 | 91.6 | 96.6 | 55.2 | 86.5 | 93.7 | 492.8 |
| VSRN(Li et al., 2019) | 76.2 | 94.8 | 98.2 | 62.8 | 89.7 | 95.1 | 516.8 |
| IMRAM(Chen et al., 2020) | 76.7 | **95.6** | **98.5** | 61.7 | 89.1 | 95.0 | 516.6 |
| OSCAR (Fine-tuned)(Li et al., 2020) | 89.8 | 98.8 | 99.7 | 78.2 | 95.8 | 98.3 | 563.6 |
| **LILE** | **77.7** | 95.4 | 98.3 | **64.1** | **91.0** | **96.4** | **522.9** |
| 5K | | | | | | | |
| SCO(Huang et al., 2018) | 42.8 | 72.3 | 83.0 | 33.1 | 62.9 | 75.5 | 369.6 |
| SCAN(Lee et al., 2018) | 50.4 | 82.2 | 90.0 | 38.6 | 69.3 | 80.4 | 410.9 |
| PVSE(Song and Soleymani, 2019) | 45.2 | 74.3 | 84.5 | 32.4 | 63.0 | 75.0 | 374.4 |
| VSRN(Li et al., 2019) | 53.0 | 81.1 | 89.4 | 40.5 | 70.6 | 81.1 | 415.7 |
| IMRAM(Chen et al., 2020) | 53.7 | **83.2** | 91.0 | 39.7 | 69.1 | 79.8 | 416.5 |
| ALIGN (Fine-tuned) (Jia et al., 2021) | 77.0 | 93.5 | 96.9 | 59.9 | 83.3 | 89.8 | 500.4 |
| OSCAR (Fine-tuned)(Li et al., 2020) | 73.5 | 92.2 | 96.0 | 57.5 | 82.8 | 89.8 | 491.8 |
| CLIP(Radford et al., 2021) | 58.4 | 81.5 | 88.1 | 37.8 | 62.4 | 72.2 | 400.4 |
| **LILE** | **55.6** | 82.4 | **91.0** | **41.5** | **72.1** | **82.2** | **424.8** |

The results are shown in Table 1 and Table 2 for MS-COCO and ARCH datasets respectively. The proposed method, called LILE, outperformed the previous best model, i.e., IMRAM (Chen et al., 2020), by a large margin of 6.3 and 8.3 in terms of overall performance R@sum in MS-COCO(1K) and MS-COCO(5K) datasets, respectively.

Table 2: Comparison with previous methods using the ARCH dataset

| Method | Text Retrieval | | | Image Retrieval | | | R@sum |
|---|---|---|---|---|---|---|---|
| | R@1 | R@5 | R@10 | R@1 | R@5 | R@10 | |
| PVSE | 36.2 | 48.8 | 53.3 | 29.3 | 43.8 | 52.6 | 264.0 |
| IMRAM | 40.3 | 52.5 | 59.5 | 32.4 | 48.2 | 56.1 | 289.0 |
| CLIP (Zero-shot) | 38.2 | 51.1 | 57.0 | 30.8 | 46.1 | 53.2 | 276.4 |
| **LILE** | **44.8** | **57.4** | **64.4** | **36.7** | **52.2** | **61.7** | **317.1** |

Because there is no previously reported results on the ARCH dataset, publicly available methods like IMRAM (Chen et al., 2020), PVSE (Song and Soleymani, 2019) and CLIP (Radford et al., 2021) were selected and modified to evaluate them on the ARCH dataset. LILE significantly outperformed the best previous best model, i.e., IMRAM, with a large margin of 28.1 in terms of overall performance R@sum. These results demonstrate the effectiveness of the proposed approach for the cross-modality retrieval task. The implementation details described in appendix C.

The cross-modal retrieval task can benefit greatly from the use of large amounts of collected data and powerful computational resources. Consequently, methods trained on large-scale datasets (i.e., CLIP (Radford et al., 2019), ALIGN (Jia et al., 2021), and OSCAR (Li et al., 2020)) can not be directly compared to methods other methods and LILE which trained exclusively on smaller datasets such as MS-COCO. The values for those methods trained on massive datasets are reported to highlight latter point.

Additionally, the results on ARCH dataset show even more improvement compared to MS-COCO results. One of the reasons that may have influenced this enhancement could be the use of Transformer for encoding the input data as IMRAM (Song and Soleymani, 2019) and PVSE (Song and Soleymani, 2019) applied GRU for their text encoding. Moreover, analyzing the results on ARCH dataset can prove that methods like CLIP (Radford et al., 2021) which have been trained on massively large datasets cannot perform very well on tasks like image-text retrieval in pathology. This type of tasks are specialized and apparently need more sophisticated methods like the proposed framework.

## 5. Conclusion

The proposed cross-modality approach can simultaneously capture the most salient features of each modality in relation to itself and other modalities. A multi-head self-attention module was used to aid the network in gaining a better understanding of each modality. Meanwhile, to assist the model in identifying all potential alignments between two modalities, the multi-head self-attention module output is fed into a cross-attention module. This approach can aid the model in adjusting retrieved features from one modality considering the effect of the other one. Additionally, the suggested novel loss objective can assist the model in determining the optimal weight to balance both implicit and explicit sources of information used to match paired instances in different modalities. Experiments on the MS-COCO and ARCH datasets demonstrated the efficiency of LILE in terms of recall, a typical metric for retrieval tasks in both general-purpose and histopathology contexts.

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

## Appendix A. Similarity Matrix

Figure 2 illustrates the similarity matrix. Images and text with a same subscript are paired instances which means the diagonal of the matrix should have the largest value compared to the other indices

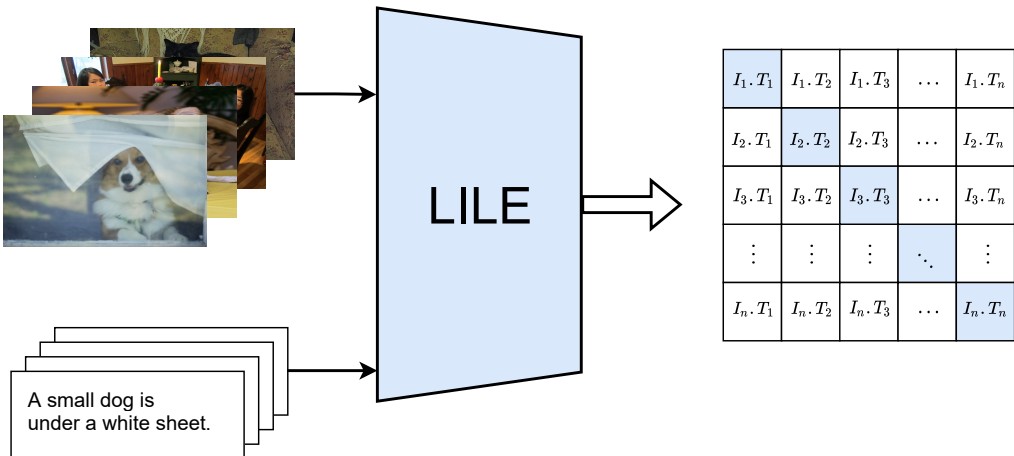

Figure 2: Similarity matrix for a mini-batch with size of $n$. Element $I_i.T_{.j}$ in the matrix shows the similarity score between image $I_i$ and text $T_j$. As the paired data has a same subscript, similarity matrix should be diagonal.

## Appendix B. ARCH dataset

ARCH dataset(Gamper and Rajpoot, 2021) is a computational pathology multiple-instance captioning dataset that includes morphological descriptions and diagnoses for a wide variety of tissue types and staining. Some samples from the dataset are shown in Figure 3. The dataset images and corresponding descriptions were mined from PubMed medical articles and pathology text books. It contains 7,579 images and a description for each image. For experiments conducted using this dataset, the 5-fold cross validation was applied.

## Appendix C. Implementation Details

For those experiments using ARCH dataset, DeiT architecture (Touvron et al., 2021) is used as the image encoder. This architecture includes 12 layers and 768 hidden state and split the input image into $16 \times 16$ patches. As the result, each input image can be seen as 196 patches with the size of $16 \times 16$. DeiT model individually is trained for a classification task on the dataset that had been used in "KimiaNet" (Riasatian et al., 2021). In the next step, the weights for the image encoder are initialized from the trained model in the previous step. As the size of the dataset was not large enough to properly train the network, only the

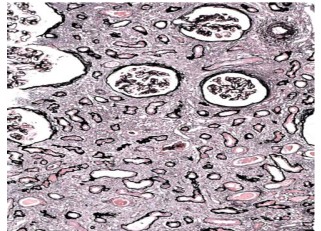
Cortex in a renal biopsy specimen from a man of 38 known to have membranous nephropathy, shown by a biopsy nearly 2 years before this one. The nephrotic syndrome had persisted since the first biopsy, and acute renal failure had developed 1 month before this biopsy. There is now widespread uniform tubular atrophy, which suggests longstanding renal vein thrombosis. This was confirmed by radiologic investigation. Renal function did not recover

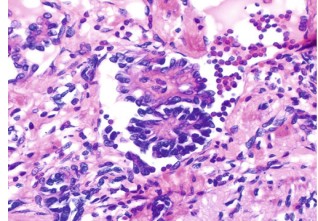
Papillary intralymphatic angioendothelioma. This is a dermal or subcutaneous lesion of children or adults, with intravascular growth of cells that have a lymphatic endothelial immunophenotype. Some have adjacent lymphangiomas or clusters of lymphatic vessels.The earlier terminology was malignant endovascular papillary angioendothelioma of childhood , but in a more recent series, reported cases did not recur or metastasize

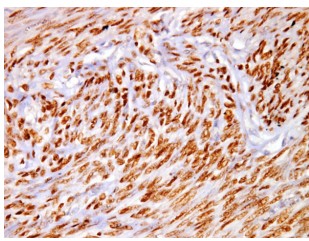
High PR expression (3+) in a leiomyoma,(PR x400)

Figure 3: Three samples from the ARCH dataset. As it can be seen, images are from different staining and and normalizations with varied description structure.

the last two blocks of image encoder were trainable and the rest of the network were frozen. For the text encoder, the network weights are initialized from the BioMed-RoBERTa-base (Gururangan et al., 2020) which had been trained on 2.68 million scientific papers from Semantic scholar corpus (Sem). This model is a language model based on RoBERTa-base architecture (Liu et al., 2019) which includes 12, 768 hidden dimensions with 110M learnable parameters. Same as image encoder, only 2 last layers of text encoder were trainable and the rest of the network were frozen. The challenging part for ARCH dataset as it shown in Figure 4 is related to the descriptions length. The descriptions are mostly non-uniform, i.e., ranging from 2 tokens to 484 tokens and much longer compared to MS-COCO dataset. To address this problem, each description is split into sentences. Then, concatenation of the first sentence with every other sentence are considered as new individual descriptions. By doing this, each image can have one or more than one description which help to augment the data and uniform the descriptions.

For experiments using MS-COCO dataset, the same architectures are implemented for both image encoder and text encoder. The only differences are the weight initialization applied for these networks and the number of layers that set to be trainable. The image encoder weights are initialized from a pre-trained model that trained on a classification task on ImageNet-1k dataset (Deng et al., 2009), which has 1 million images with 1,000 classes. Moreover, the input data for image encoder is the feature maps that had been extracted using Faster R-CNN as explained in section method section. The last 6 layers of the image encoder were trainable and the rest were frozen. The text encoder weights are initialized from a pre-trained Roberta-base network that had been trained on the reunion of five general datasets[1]. The last 6 layers of the text encoder were trainable and the rest were frozen.

All the experiments are implemented in Pytorch V1.3 and run with two NVIDIA V100 GPUs. For the conducted experiments, images were resized to $224 \times 224$ pixels and normal-

---

1. BookCorpus (Hom), which contains 11,038 unpublished books, English Wikipedia (Eng), CC-News (New, b), which contains 63M English news articles, OpenWebText (Git), which is a WebText dataset used to train GPT-2 and Stories which is a subset of CommonCrawl (New, a) data.

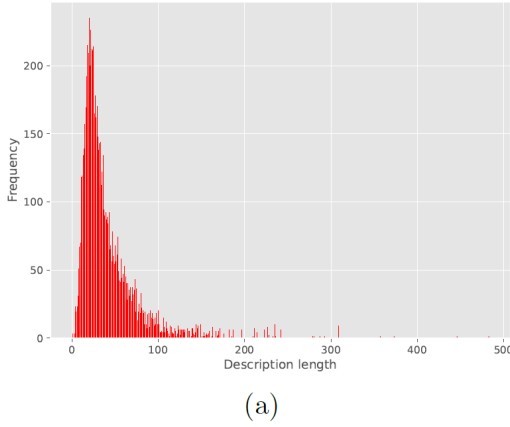
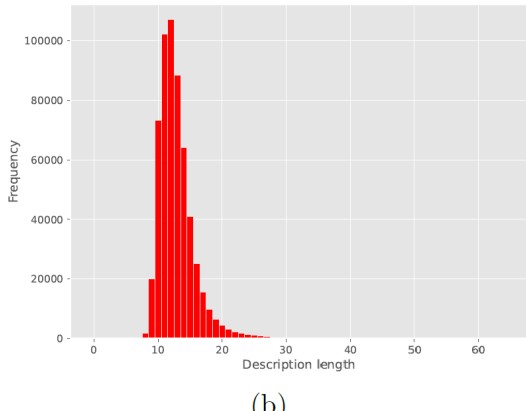

(a)                                              (b)

Figure 4: (a): Frequency of the length of the descriptions tokens in the ARCH dataset. (b): Frequency of the length of the descriptions in the MS-COCO dataset. The MS-COCO dataset has an average of 13.81 and a maximum of 65 tokens per caption, whereas The ARCH dataset has an average of 51.49 and a maximum of 484 for each description.

Table 3: The effect of the number of matching steps, $K$, in MS-COCO dataset.

| K | Text Retrieval | | | Image Retrieval | | | R@sum |
|---|---|---|---|---|---|---|---|
| | R@1 | R@5 | R@10 | R@1 | R@5 | R@10 | |
| 1 | 77.2 | 96.4 | 99.0 | 64.4 | 92.0 | 96.6 | 525.6 |
| 2 | 79.6 | 97.4 | 99.7 | 67.0 | 92.8 | 97.2 | 533.7 |
| 3 | 79.9 | 97.0 | 99.0 | 66.2 | 92.3 | 97.3 | 531.7 |
| 4 | 78.5 | 96.7 | 98.8 | 65.9 | 92.5 | 97.4 | 529.8 |

ized before feeding into the model. The hyper-parameter $m$, which specifies the number of self-attention heads, is set as 16 for optimal performance across all experiments. The Adam optimizer (Kingma and Ba, 2014) is used with an initial learning rate of $1e-4$. The learning rate is decreased when the metric has stopped improving. The margine for contrastive loss function set to be 0.2.

## Appendix D. Effect of the iterative Matching scheme, $K$

To better understand the effect of $K$ in the proposed approach, $K$ gradually is increased from 1 to 4 to train and test on the MS-COCO (5K) benchmark dataset. Table 3 shows the results for R@1, R@5, R@10 and R@sum metrics on image-to-text retrieval and back. It can be observed for $K = 2$ that the model consistently achieved better performance than other values for $K$. The gap between the performance for $K = 1$, which is a baseline without the iterative scheme, and other values for $K$ is significant. That gap can prove the importance of the iterative scheme in the proposed approach.

