# OpenReview forum: "LILE: Look In-Depth before Looking Elsewhere -- A Dual Attention Network using Transformers for Cross-Modal Information Retrieval in Histopathology Archives"
_MIDL.io/2022/Conference — MIDL 2022_

### Official Review · Reviewer_msBw · 2022-01-10

**Confidence:** 3
**Preliminary Rating:** 5
**Recommendation:** Poster

**Summary:**

In this paper, the authors propose a method for bidirectional image retrieval. They use transformer networks to extract and feature representation of both text and image data. Their approach features self-attention modules with iterative matching scheme and a custom loss function to enhance the feature representation. The authors evaluate their method on two publicly available datasets, of which one is in the pathology domain and compare the results to state-of-the-art approaches.

**Strengths:**

* The paper is well-structured and straightforward
* The datasets used in the paper are publicly available
* The method outperforms state-of-the-art on widely used MS-COCO dataset
* The approach is described in good detail

**Weaknesses:**

* It is not described how the datasets are split and whether a hold-out test set was used or whether the test set was used to optimize the approach
* Also, there are no training parameters listed such that is appears that the paper alone is not sufficient to reproduce the results
* The application of the proposed method is not described. At least some example of real-world-problem in digital pathology should be given that the proposed method solves


**Deanonymize Review:**

no

**Detailed Comments:**

* The paper could be improved by adding some more figures that illustrate the application or some exemplar results


**Paper Type:**

methodological development

**Questions To Address In The Rebuttal:**

 The authors should provide an explanation why their approach outperforms the other approaches by a small margin for MS-COCO but by a quite large margin for ARCH. Is the approach better suited for pathology data? If so, why? Or is it just because much effort was put into optimizing the own approach for ARCH while the other methods were not optimized for that particular dataset?
* further questions see weaknesses and comments

**Special Issue:**

yes

---

### Official Review · Reviewer_tibw · 2022-01-22

**Confidence:** 3
**Preliminary Rating:** 3

**Summary:**

The authors proposed a Transformer-based architecture with a new loss term to improve the representation of images and texts in the joint latent space.
Experiment results on two benchmark datasets, i.e. MS-COCO and ARCH, show the effectiveness of the proposed method in cross-modality retrieval for histopathology images.


**Strengths:**

1. The authors proposed a cross-modality method to extract the most significant features by looking into its context.
2. The authors trained the model in an iterative regime to help the model to capture higher-level features gradually based on the features
acquired in the previous steps.
3. Experiments on two public datasets were shown.

**Weaknesses:**

Technical contribution:
The technical contribution of this paper is not highlighted. Most of the components seem to be off-the-shell. Is the cross-attention module a contribution?

New loss function:
The author claimed that they proposed a new loss in the Abstract. But I could not find it in the text. Please clarify this if I miss the information.

Parameter setting:
1) What is the margin value used in the triplet loss?
2) How the datasets were pre-processed, what are the training/validation/test splits?
3) Hyperparameters such as the margin value, are not discussed.

Experiments:
1) The benefit of iterative matching is not demonstrated.
2) it would be good to show some examples for both directions: image to text, and text to image.

Minor:
Equation 6, what is 'sim'? is it the distance similarity?





**Deanonymize Review:**

no

**Detailed Comments:**

Technical contribution:
The technical contribution of this paper is not highlighted. Most of the components seem to be off-the-shell. Is the cross-attention module a contribution?

New loss function:
The author claimed that they proposed a new loss in the Abstract. But I could not find it in the text. Please clarify this if I miss the information.

Parameter setting:
1) What is the margin value used in the triplet loss?
2) How the datasets were pre-processed, what are the training/validation/test splits?
3) Hyperparameters such as the margin value, are not discussed.

Experiments:
1) The benefit of iterative matching is not demonstrated.
2) it would be good to show some examples for both directions: image to text, and text to image.

Minor:
Equation 6, what is 'sim'? is it the distance similarity?



**Final Rating After The Rebuttal:**

2: Weak Reject

**Justification Of The Final Rating:**

1. Unclear contributions.

2. Quite some places (data split, parameters setting) need clarifications as mentioned by our three reviewers.

3. Rebuttal/response is a part of the process. No response may harm the review process.




**Paper Type:**

both

**Questions To Address In The Rebuttal:**

1) the main contribution of this paper.
2) details on the pre-processing of the datasets, e.g. training/validation/test splits.
2) the discussion on parameter setting, such as the margin value.
3) the benefit of iterative matching is not demonstrated.


**Special Issue:**

no

---

### Official Review · Reviewer_EGnq · 2022-01-23

**Confidence:** 3
**Preliminary Rating:** 2
**Recommendation:** Poster

**Summary:**

The combination of multi-modal data is relevant across different fields of applications, and stretches also to the medical domain. The authors describe a method for text / image retrieval based on (mostly) transformers and using an attention-based approach. For this, they describe the combination of self-attention and cross-attention modules within a transformer architecture to enrich the representation in the latent space. Furthermore, they include an iterative matching scheme to improve matching of higher-level features. They evaluate their approach on MS COCO 1K and 5K as well as ARCH, a histopathology dataset with moderate improvements on MS COCO and a larger performance margin on ARCH.

**Strengths:**

The paper shows strength in that it acknowledges and demonstrates the wealth of work that has been done on multi-modal image retrieval: The paper features a comparatively extensive related work section on different aspects pertaining to the presented work. Furthermore, the proposed approach is compared with 3 (ARCH) - 8 (COCO 5k) approaches to assess its performance.

**Weaknesses:**

When reading the paper, it was difficult to understand the difference of the paper compared to the paper by Chen et al. from 2020, as multiple formulations are very similar, but this is not always acknowledged in the work at hand. Specifically, after reading the paper, it is not clear to this reviewer what the contribution w.r.t. "a novel architecture" and "a new loss term" is (see abstract). This should be mentioned more clearly in the introduction and in the respective parts of the methodology section. While a "self-attention module" is mentioned, this not described in more detail and for example Wei et al. [r1] could be mentioned here. The cross-module attention as well as the iterative scheme is described very similarly in Chen et al. Additionally, the mathematical formulations and the structure of the paper (see also below) don't seem to be consistent and make the paper hard to follow and the similarities and differences / contributions further difficult to identify.

[r1] Xi Wei, Tianzhu Zhang, Yan Li, Yongdong Zhang, Feng Wu; Multi-Modality Cross Attention Network for Image and Sentence Matching-Proceedings of the IEEE/CVF Conference on Computer Vision and Pattern Recognition (CVPR), 2020, pp. 10941-10950

**Deanonymize Review:**

yes

**Detailed Comments:**

Methodology & contribution:
- The methodology section is relatively complex to read as it is not clear which part will be described when, what is described later on in more detail and what should be taken as known modules (e.g., iterative matching scheme, gated memory block, cross attention module). This could potentially be clarified by explaining the structure of the methodology section at the beginning or end of the first paragraph. E.g., why is the first subsection "Feature representation"? A reader has to match these parts with the previous description going repeatedly back and forth between the sections. Why is section 4 (iterative matching) not a part of the methodology?
- The authors introduce object detection models as an alternative means of representing image data in section 3.1 - this is rather confusing since this aspect has not been mentioned before, but instead, emphasis was put on transformers as a means of extracting a data representation.
- In the introduction and in the methodology section, the authors could delineate more clearly what their contribution is (feature representation, attention modules, gated memory blocks, and the iterative matching scheme (see above)

Section 4: The section on iterative matching was not clear to this reviewer. (see above)
- It seems to closely follow the description in Chen et al. 2020, but this is not acknowledged nor are potential differences made clear. This applies both to the iterative matching itself and also to the loss function.
- The equations in this section don't seem to be well connected with the previous section, e.g., X*_k, Memory are not used in the remainder of the text, X_{k-1} is not clearly defined, ...
- The connection of Eq. 4 to the remainder of the text is not clear.
- For Eq. 5/6, the image regions and text tokens should be clearly described by their variables (simply mention v_i and w_i again). (see also above)
- The function "sim" is not clear - I would assume that this relates to the previous section and the similarity defined here? How are the direct similarity scores computed? How is sim() is computed once for the regions/tokens and once for the full image/text?

Results:
- A short description of how the retrieval task is performed and evaluated on a captioning dataset (ARCH) would make the evaluation more clear, especially since the captions in ARCH are partially quite complex.
- Why are ALIGN and CLIP not included in the 1K comparison? How where the publicly available methods trained / adapted on ARCH? What does (zero-shot) mean for CLIP? Wouldn't it make sense to also adapt this?
- Since the ARCH paper also proposes a baseline algorithm, the authors should at least shortly mention why they don't compare themselves with the baseline.
- What is the variability of the results for repeated trainings on MS-COCO/ARCH resp. for the 5-fold cross-validation? What does a 1.0 percentage point difference mean for this dataset? A short discussion of that would be helpful.

- The authors have included a 3 page appendix, but from the paper itself, it is not clear for which parts the reader should look into the appendix for additional information. This could shortly mentioned within the respective section of the main paper.

Minor comments:
- "Furthermore, the age of networks that used multiple modalities separately has practically ended." - this seems to be an overly general statement
- "However, regardless of their importance in their own modality, these approaches usually consider features of each modality equally." - what does this mean?
- Although commonly used in the transformer architecture context, a short definition and / or a reference that hosts a definition of "self-attention" and "cross-attention" could clarify the meaning with which the authors use these two terms and illustrate the difference better.
- Generally, there are some issues with expression, and additional proof-reading is recommended, a few examples are listed below (not exhaustive).
- pg. 1: "However, The" - typo
- pg. 1: "The diversity in information sources can be observed more commonly in the medical field. [...] A model able to retrieve a source of data when a different one is available can help provide complete information about the patient." - what do the authors want to say here? This is a bit unclear.
- pg. 2: "The common solution to approach to explore" - expression
- pg. 2: The use of "modal" as a noun strikes me as very uncommon, and I remember this being used only as an adjective in related work. Here, modality may be the better choice.
- pg. 3: The authors repeatedly use the first name with et al., e.g., Yang et al. / Christopher et al. , it is also uncommon to use "et al." for two authors.
- pg. 3: "in IMRAM paper" - expression - occurs multiple times
- pg. 3: "task of image-text retrieval could be outperformed using a massive dataset" - expression
- pg. 3: "such as histopathology domain"
- pg. 3: "Cross-Modality Retrieval in is new"
- pg. 3: "Yet the developed dataset was too small" - too small for what?
- Fig. 1: the text in the figure is extremely small
- pg. 4: "using transformers [...] using transformers" - there are generally multiple rather repetitive formulations & mistakes in this section.
- pg. 4: "Transform" vs. "Transformers"



**Final Rating After The Rebuttal:**

3: Borderline

**Justification Of The Final Rating:**

The authors commented very late on the paper, which did not allow for a more detailed discussion of their paper and their contributions, and left little time for a final assessment.
It is appreciated that the authors detailed parts of their method and contributions a bit more clearly in the revised version; however, from my perspective the paper still requires a very alert reader.

The image-retrieval aspect is certainly interesting to the MIDL community, which leaves me between a borderline and a weak accept.

**Paper Type:**

both

**Questions To Address In The Rebuttal:**

Contribution:
- What do the authors see as the main contribution especially compared the Chen et al. 2020, and to the aforementioned publication by Wei et al.? Looking at the descriptions in Chen et al., a difference seems to be the explicit mentioning of a GRU for the gated memory unit, but the authors also use a gated mechanism which they don't explain further.

Applying image retrieval to ARCH does not seem to be fully straightforward:
- How was the image retrieval task performed / evaluated given the partially very long captions in the ARCH data set?
- How where the reference methods trained / optimized?
- How did the authors deal with the multi-instance setting within ARCH?

**Special Issue:**

no

---

### Meta-Review · Area_Chair_uz85 · 2022-02-20

**Recommendation:** Accept (Poster)
**Confidence:** 2

**Metareview:**

The authors claim that the proposed method is the first fully bi-directional cross-modality retrieval in the field of histopathology. At the same time, some reviewers initially pointed out the lack of novelty of this work, which was then partly clarified during the rebuttal phase, at least with reviewer 1.

---

### Decision · Program_Chairs · 2022-02-28

Accept